# Cationic Cellulose Nanocrystals-Based Nanocomposite Hydrogels: Achieving 3D Printable Capacitive Sensors with High Transparency and Mechanical Strength

**DOI:** 10.3390/polym13050688

**Published:** 2021-02-25

**Authors:** Po-Cheng Lai, Sheng-Sheng Yu

**Affiliations:** Department of Chemical Engineering, National Cheng Kung University, No. 1 University Road, Tainan 70101, Taiwan; N36094136@gs.ncku.edu.tw

**Keywords:** 3D printing, nanocomposite hydrogels, cellulose nanocrystals, stretchable electronics

## Abstract

Hydrogel ionotronics are intriguing soft materials that have been applied in wearable electronics and artificial muscles. These applications often require the hydrogels to be tough, transparent, and 3D printable. Renewable materials like cellulose nanocrystals (CNCs) with tunable surface chemistry provide a means to prepare tough nanocomposite hydrogels. Here, we designed ink for 3D printable sensors with cationic cellulose nanocrystals (CCNCs) and zwitterionic hydrogels. CCNCs were first dispersed in an aqueous solution of monomers to prepare the ink with a reversible physical network. Subsequent photopolymerization and the introduction of Al^3+^ ion led to strong hydrogels with multiple physical cross-links. When compared to the hydrogels using conventional CNCs, CCNCs formed a stronger physical network in water that greatly reduced the concentration of nanocrystals needed for reinforcing and 3D printing. In addition, the low concentration of nanofillers enhanced the transparency of the hydrogels for wearable electronics. We then assembled the CCNC-reinforced nanocomposite hydrogels with stretchable dielectrics into capacitive sensors for the monitoring of various human activities. 3D printing further enabled a facile design of tactile sensors with enhanced sensitivity. By harnessing the surface chemistry of the nanocrystals, our nanocomposite hydrogels simultaneously achieved good mechanical strength, high transparency, and 3D printability.

## 1. Introduction

Hydrogel ionotronics, made of a material with high stretchability and high ionic conductivity, have received much attention for their applications in wearable sensors [1], soft robots [2], and artificial muscles [3]. For most hydrogels, water molecules are trapped in macromolecules to enable the transportation of mobile ions [4]. Stretching or compressing hydrogels leads to a change of resistance. Hydrogels may also be assembled with dielectrics to provide changes of capacitance during mechanical deformation [5]. Highly transparent hydrogels further allow the transmittance of visible content [6], such as electroluminescence [7] inside the devices. Besides conducting polymers [8] and inorganic nanocomposites [9], hydrogels are popular materials for soft electronics that monitor various human activities. However, the polymeric networks inside the hydrogel ionotronics have to be tailored to achieve high mechanical strength, high ionic conductivity, and good transparency. Furthermore, the applications of hydrogel ionotronics may be expanded if additive manufacturing can be used to fabricate functional structures.

Besides the single covalently cross-linked polymers, several strategies have been proposed to design hydrogels. For example, tough hydrogels can be achieved by using double covalent networks that utilize the fracture of stiff networks and the subsequent dissipation of energy [10]. The combination of covalent and physical networks also enhances the stiffness and toughness of hydrogels [11,12]. Sun et al. used the strong ionic association of polyampholytes to produce tough and viscoelastic hydrogels [13]. Furthermore, fillers with high mechanical strength, such as clay [14] and SiO_2_ [15], can be introduced to synthesize reinforced nanocomposite hydrogels. The association between polymer chains and the surface functional groups of the nanofillers gives an additional dimension to control the mechanical properties of the hydrogels [16]. Finally, the recently developed zwitterionic polyelectrolyte provides a simple strategy to make hydrogels with a wide spectrum of properties [17]. While the charge balance of polyampholyte-based hydrogels has to be precisely controlled, the composition of monomers in zwitterionic hydrogels can be easily tuned for different degree of hydrogen bonding, ionic association, and mechanical properties.

The development of additive manufacturing or 3D printing has shown great potential for rapid prototyping and customization. 3D printing enables the precise construction of complex geometry, which might be difficult for the traditional molding method. Among the many 3D printing techniques, direct ink writing (DIW) is a simple and versatile method to process a variety of materials. DIW often utilizes the viscoelasticity of non-Newtonian fluids from colloidal gels [18], micellar gels [19], or physically cross-linked biopolymers [20]. The shear-thinning behavior of the ink allows fast extrusion of materials under mild pressure. After extrusion, the physical network inside the ink rapidly reforms to retain the printed structures [21]. The versatility of DIW facilitates the rapid prototyping of different materials for applications like tissue engineering [22], wearable electronics [23], and soft robots [2]. For example, Yuk et al. recently used DIW to print the conducting polymers poly(3,4-ethylenedioxythiophene):polystyrene sulfonate [24]. Multi-material 3D printing enables the rapid fabrication of neural probes by continuously printing insulating layers and the conducting polymers. Wei et al. used nanoclay as the rheological modifier to prepare nanocomposite ink based on thermoplastic polyurethane and carbon black [25]. Their printed object further formed porous structures for strain and gas sensing.

For the reinforcement of hydrogels and the design of ink for DIW, the fillers play an important role. Cellulose nanocrystals (CNCs) are sustainable materials that can be extracted from biomass by acid hydrolysis [26]. CNCs are lightweight nanorods with high crystallinity due to the close packing of cellulose polymers. Strong polymer nanocomposites [27], hydrogels [28], and aerogels [29,30] have been developed by using CNCs. In addition, CNCs are excellent rheological modifiers for DIW. When dispersing in water, CNCs form percolated networks through the balance between van der Waals forces and electrostatic repulsion [31]. Although other cellulose-based materials such as methylcellulose [32,33] have also been used for 3D printing, CNCs offer advantages for the structuring of designs by DIW. For example, the shear stress during the extrusion process can further align the CNCs along the printing direction to achieve anisotropic mechanical reinforcement and optical properties [34,35]. However, a high concentration of CNCs is often used for strong nanocomposites. It leads to the difficulty of dispersing the nanocrystals and the opaqueness of the final product. The surface chemistry of the nanocrystals has to be properly engineered to address the above challenges. CNCs have abundant surface hydroxy groups for chemical modification. For example, CNCs may undergo TEMPO-oxidation that converts the hydroxy groups into carboxylic groups [36]. Positive charges can also be introduced by the reaction of CNCs with epoxy propyltrimethylammonium chloride (EPTMAC) [37]. The surface chemistry has a significant effect on the rheological behavior of CNCs dispersion [31] and polymer–filler interaction [38]. Besides the type of polymer networks, the surface chemistry of the nanofillers is another design principle for tough nanocomposites.

Our previous work [39] used unmodified CNCs to prepare ink for 3D-printable strain sensors. Although we achieved gels with high mechanical strength and stretchability, the materials were not transparent due to the high particle loading. To address this challenge, we investigated the 3D printing of zwitterionic nanocomposite hydrogels with modified CNCs. Our design used cationic cellulose nanocrystals (CCNCs) for DIW, as shown in Figure 1. The ink was composed of CCNCs, acrylic acid (AA), and 3-dimethyl (methacryloyloxyethyl) ammonium propane sulfonate (DMAPS). After printing, the zwitterionic nanocomposite hydrogels with multiple physical cross-links were formed by the photo-initiated polymerization of AA/DMAPS, and the introduction of Al^3+^ ions. Although some previous studies have attempted to use CCNCs for nanocomposite hydrogels, the potential of using such CCNC-based hydrogels for 3D printing and wearable sensors has not been explored yet [40,41]. We found that CCNCs had a stronger tendency to form physical networks than conventional CNCs. Therefore, the nanocomposite hydrogels of CCNCs could be printed with a low concentration of fillers to achieve high transparency but still demonstrated good mechanical strength. Finally, we assembled the CCNC-reinforced zwitterionic nanocomposite hydrogels as a capacitive sensor capable of monitoring different mechanical deformations. We demonstrated that DIW could print hydrogels with a grooved surface to enhance the sensitivity of tactile sensors.

## 2. Materials and Methods

### 2.1. Materials

The acrylic acid (AA, 99.5%, stabilized with 200 ppm of 4-methoxyphenol) and aluminum chloride hexahydrate (AlCl_3_
**·** 6H_2_O, 99%) used were from Alfa Aesar, Ward Hill, MA, USA. 3-dimethyl (methacryloyloxyethyl) ammonium propane sulfonate (DMAPS, >98%), 2-hydroxy-4′-(2-hydroxyethoxy)-2-methylpropiophenone (Irgacure 2959), and epoxy-propyltrimethylammonium chloride (EPTMAC) were obtained from TCI, Tokyo, Japan. Sodium hydroxide (NaOH) was from UniRegion, New Taipei City, Taiwan. Sodium chloride (NaCl) was from SHOWA, Tokyo, Japan. CNCs were purchased from Celluforce Inc., Montreal, QC, Canada. The length of the CNCs ranged from 44 to 108 nm and the diameter was between 2.3 and 4.5 nm, as determined by atomic force microscopy (Bruker, Billerica, MA, USA).

### 2.2. Synthesis of CCNCs

For the modification of CNCs, we followed the process described by Hasani et al. [37] First, 20 g of CNCs was added into 200 mL of deionized water, then 100 g of 20 wt.% NaOH solution was introduced into the dispersion under strong mechanical stirring for 30 min. EPTMAC was then dropped into the dispersion. The mixture was further stirred at 65 °C for 6 h. After the reaction, the solution was diluted twofold with water. The suspension was then centrifugated at 2000 rcf for 10 min and washed three times with water. The solution was dialyzed for several days until the solution became neutral. Finally, the dispersion of CCNCs was lyophilized for further use.

### 2.3. Preparation of the CCNC Nanocomposite Gels

AA and DMAPS (weight ratio of 4:1) were mixed in deionized water to obtain an aqueous solution with a concentration of 45 wt.% monomers. Irgacure 2959 (0.5 mol% of monomers) was added as a photoinitiator. CNCs or CCNCs (0 to 12 wt.%) were dispersed in the aqueous mixtures of monomers with a speed mixer (MV-300S, CGT, New Taipei City, Taiwan) at 1500 rpm for 10 min with four Teflon balls (diameters: 1.27 and 0.64 mm). The obtained dispersion was used for 3D printing and rheological analysis. For mechanical and conductivity tests, the ink was photopolymerized by UV light of 365 nm (20 mW/cm^2^) for 30 min inside a rectangle mold. The polymerized hydrogels were then immersed in a 0.5 M aqueous solution of AlCl_3_ for 10 h to introduce the ionic cross-links. All samples for mechanical and electrical measurement were immediately used after preparation.

### 2.4. Rheological Analysis

The rheological experiment for each sample was done by an HR-2 rheometer (TA Instruments, New Castle, DE, USA) with a parallel plate, 25 mm in diameter. The shear storage modulus and shear loss modulus were measured using an oscillatory stress sweep at a frequency of 1 Hz at room temperature.

### 2.5. Mechanical Test

Mechanical properties of the samples were measured using a universal testing machine (Shimadzu AGS-X, Shimadzu, Kyoto, Japan) with a 100 N load cell. The hydrogels were cut into dumbbell shapes based on the ASTM-D638-type V standard. The crosshead speed was 50 mm/min during the tensile tests. For the cyclic tensile tests, samples were loaded at 50 mm/min and then unloaded at the same rate. The amount of dissipated energy (*U_hys_*) of each sample was calculated by the area between loading–unloading curves.

### 2.6. 3D Printing

The inks were loaded into a 10 mL syringe and centrifuged at 1500 rpm for 1 min to remove air bubbles. 3D printing was performed using a fused deposition modeling (FDM) printer (INFINITY X1Speed, INFINITY3DP, Kaohsiung, Taiwan). The extruder was replaced with a 3D printed adapter for DIW. The nozzle (0.4 mm in diameter) was mounted onto the adapter and connected to a syringe pump (NE-1000, New Era, Farmingdale, NY, USA). The initial distance of the nozzle tip from the platform was 0.4 mm. The printer was controlled using Repetier software (Hot-World GmbH & Co. KG, Willich, Germany). The typical printing speed was 20 mm/s. The printed gels were then cured in a chamber without deoxygenation using a UV light source (365 nm, 20 mW/cm^2^) above the sample for 30 min.

### 2.7. Electrical Measurement

The ionic conductivities of the nanocomposite hydrogel were measured by electrical impedance spectroscopy (EIS) with a potentiostat (PGSTAT302, Autolab, Eco-Chemie, Utrecht, The Netherlands). The cured nanocomposite hydrogels were cut into disks of 1 and 0.2 cm in diameter and thickness, respectively. The disks were then sandwiched between two stainless-steel electrodes for the EIS measurement, with a frequency range from 10^5^ to 10^−1^ Hz and a disturbance voltage of 10 mV. The resistance of the hydrogel, *R* (Ω), was found by calculating the intercept between the impedance curve and the real axis (*Z’*). The ionic conductivity, *σ* (S/m), was then calculated by *σ* = *d*/(*R* × *S*), where *d* is the thickness of the disk and *S* is the interfacial area between the disk and the electrode.

The real-time capacitive change in the nanocomposite hydrogels was measured using a multimeter (DMM6500, Keithley, Solon, OH, USA). For the fabrication strain sensors, VHB tape (3 M, thickness 0.5 mm) was used as the dielectrics. The VHB tape was sandwiched between two 3D-printed hydrogels (5 cm in length, 2 cm in width, and 1.2 cm in thickness). The samples were first connected to the multimeter by aluminum foils, mounted to the universal testing machine, and then stretched at a strain rate of 1000 mm/min. The relative change in capacitance at different strains was calculated by 100% × (*C* − *C_0_*)/*C_0_*. *C* is the capacitance during stretching and *C_0_* is the initial capacitance. For tactile sensors, the hydrogels (3 cm in both length and width) were 3D printed based on the design shown in Appendix A. The tactile sensors were compressed by the universal testing machine at a rate of 0.5 mm/min to determine the gauge factors.

### 2.8. Characterization

The zeta potentials of CNCs and CCNCs were analyzed by Malvern Zetasizer Nano S (Malvern, UK). Concentrations of 0.5 wt.% of CNC and CCNC dispersions were used for the measurements of zeta potentials. Elemental analysis of the nanocrystals was done with an elemental analyzer (Elementar UNICUBE, Langenselbold, Germany). Solid-state ^13^C NMR spectra of CNCs and CCNCs were recorded on a Bruker Avance III HD 400 MHz NMR spectrometer (Bruker, Billerica, MA, USA) with a 4 mm probe. The magic angle spinning rate was 12 kHz. The size distributions of CNCs and CCNCs were measured by atomic force microscopy (AFM). Diluted aqueous solutions of the nanocrystals were dropped on freshly cleaved mica for 2 to 3 min, washed with water, and dried. The AFM images were obtained by a Bruker Dimension Icon (Bruker, Billerica, MA, USA) in tapping mode (Olympus OMCL-AC160TS tip, Olympus, Tokyo, Japan). The hydrogels were immersed in liquid nitrogen and lyophilized for Fourier transform infrared spectroscopy (FT-IR) and scanning electron microscopy (SEM). IR data of the samples were obtained using a Nicolet 6700 Fourier transform infrared spectrometer (Thermo Scientific, Waltham, MA, USA) and an attenuated total reflectance (ATR) sample chamber. Spectra were recorded in a range of 650–4500 cm^−1^ with a resolution of 2 cm^−1^ and 32 scans. SEM was performed by mounting the samples on a stainless-steel holder with copper tape. The images were taken by a scanning electron microscope (Hitachi SU8010, Hitachi, Tokyo, Japan) at an accelerating voltage of 10 kV and a distance of 11.2 mm. The transmittances of the hydrogels were recorded by a Hitachi U4100 UV-vis spectrophotometer (Hitachi, Tokyo, Japan). Hydrogels were prepared as rectangular films of 5 mm in thickness. UV-vis spectra were recorded in a range of 300–800 nm. The gravimetric method was used to measure the water contents of hydrogels. First, the weight of each hydrogel (*m_wet_*) was recorded and then the hydrogel was lyophilized to obtain the dry weight (*m_dry_*). The water content can be calculated by 100% × (*m_we_*_t_ − *m_dry_*)/*m_we_*_t_.

## 3. Results

### 3.1. Rheological Behavior of the CCNC Dispersion in Water

First, we investigated the effect of surface charges on the rheological behavior of the ink prepared from CNCs. CCNCs were synthesized through the reaction of EPTMAC with commercially available CNCs. According to the zeta potential analysis (Appendix A), CNCs before modification exhibited negative surface charges (−75.2 ± 1.7 mV) due to the sulfate half-ester groups, retained from the sulfuric acid hydrolysis. The zeta potential of the CCNCs was +50.8 ± 1.2 mV, indicating a successful modification. The elemental analysis of the nanocrystals also suggested an increase in nitrogen content after the modification (Appendix A). Furthermore, solid-state ^13^C NMR spectra of the nanocrystals (Appendix A) showed new signals for the methyl groups of EPTMAC in CCNCs [42]. As shown by Hasani et al., this process has a minimal effect on the size distribution of the nanocrystals [37]. In Appendix A, the length and width of the nanocrystals were revealed by height images of atomic force microscopy (AFM). The size of the native CNCs was comparable to the published results [43]. The AFM images also showed that the size distribution of CCNCs was similar to CNCs. Therefore, the effect of CCNCs on the rheological behavior of the ink and the mechanical properties of the nanocomposite hydrogels should have mainly come from the surface charge of the nanocrystals.

We then compared the rheological behavior of the ink with native CNCs and CCNCs. The inks were prepared by dispersing different concentrations of nanocrystals in the aqueous solution of AA and DMAPS. As indicated in Figure 2a,d, the inks of both CNCs and CCNCs showed shear-thinning behavior. When the shear rate increased from 0.1 to 100 1/s, the viscosity of the ink containing 7 wt.% CCNCs decreased significantly from 2515 Pa∙s to 1.8 Pa∙s. Notably, the inks with CCNCs were more viscous than the ones prepared from CNCs. Under the same filler content (7 wt.%), the viscosity of the CCNC ink at 0.1 1/s was 15 times higher than that of the CNCs. At least 12 wt.% CNCs was needed to achieve the viscosity of the 7 wt.% CCNC ink. We also attempted to use the power-law model to analyze the shear-thinning behavior of the inks for 3D printing (Appendix A) [44,45]. The flow indexes (n) of both inks were less than 1 at low filler contents. Similar to the previous studies of concentrated CNCs suspension, the flow indexes of CNC and CCNC inks were close to zero, indicating the formation of yield stress fluids [46].

We also used oscillatory stress-sweep measurements to understand the strength of the physical network formed by the nanocrystals. As shown in Figure 2b,e, both of the types of ink formed percolated networks and exhibited well-defined yield stresses (*G’* = *G*”). Figure 2c,f summarize the findings for the *G’_0_* and yield stress with different filler contents. Similar to the results of the steady-state shear viscosity, CCNCs exhibited higher initial storage *G’_0_* and yield stress than CNCs under the same particle loading. The storage modulus shows the strength of the percolated network formed by the nanocrystals. From the above results, CCNCs showed a higher tendency to gelation than CNCs. The network of CCNCs was also stronger than the one formed by CNCs under the same filler content. Similar behavior was also reported by Hasani and Tang [37,47]. In general, the colloidal stability and the aggregation behavior of the rod-like nanocrystals depend on the balance of attractive van der Waals forces and the repulsive electrostatic interaction. As discussed earlier, the CCNCs had a lower magnitude of zeta potential than the native CNCs. The decrease of surface charge density weakened the repulsive force and promoted the gelation of the nanocrystals. Therefore, CCNCs had a more significant effect on the rheological behavior of the ink than CNCs. On the other hand, the addition of acids or salts increased the ionic strength of the ink. High ionic strength leads to strong physical hydrogels due to the decreased repulsive force [46]. We anticipated that the monomers, AA and DMAPS, would increase the ionic strength of the ink to further promote the gelation of both CNC and CCNC inks.

For DIW, the ink should display a shear-thinning behavior to promote material extrusion [18]. Also, a high storage modulus and yield stress are required to ensure good shape fidelity during printing [48]. The above criteria could be achieved by using 7 wt.% CCNCs. On the other hand, the concentration of CNCs had to be at least 12 wt.% for printing. It is possible to design the ink of CCNCs with an even higher concentration. However, we found that dispersing CCNCs at concentrations higher than 7 wt.% was difficult due to the high viscosity and the strong tendency for gelation. Since the ink with 7 wt.% CCNCs already presented an ideal rheological behavior for DIW, this type of ink was primarily used for our later printing tests and the design of capacitive sensors.

From the above rheological analysis, we found that the inks using 12 wt.% CNCs or 7 wt.% CCNCs were most suitable for the DIW process. Both inks had high storage modulus, high yield stress, and prominent shear-thinning behavior. If the concentration of CNCs was less than 12 wt.%, the gels were too weak to retain the printed structures. In addition, it was difficult to properly disperse highly concentrated CCNCs in the ink. Therefore, we performed 3D printing tests using the inks with 12 wt.% CNCs and 7 wt.% CCNCs to compare the optical properties of the printable hydrogels (Figure 3a). The printed structures were solidified by UV curing of AA and DMAPS. Both inks could be extruded smoothly and stacked successfully. The structures printed by the CNC ink were more opaque than those using the CCNC ink. As shown in Figure 3b, the hydrogels without any nanocrystals had 88% transmittance at 800 nm. Gradually increasing the amount of CNCs or CCNCs reduced the transmittance. The nanocomposite hydrogels with 7 wt.% CCNCs exhibited high transmittance, up to 72% at 800 nm. On the other hand, the maximum transmittance of the 12 wt.% CNC nanocomposite hydrogels was only 48% at 800 nm. Even at the same concentration of nanocrystals (5 wt.%), the CCNC hydrogels showed higher transmittance than the CNC hydrogels. The opacity of the hydrogels comes from the high particle loading and the aggregation of the nanocrystals, which are common for nanocomposites of cellulose nanocrystals. Based on the above results, we concluded that the aggregated bundles of CNCs should be larger than those of the CCNCs [49]. Although both inks could be successfully printed, the high transmittance of the CCNC inks provides greater potential than CNCs for wearable electronics and smart displays, as high transparency is often desired [50].

### 3.2. Mechanical Properties of the CCNC Nanocomposite Hydrogels

Although printable ink can be prepared by using a low concentration of CCNCs, the decreased concentration of nanofiller may lead to weak hydrogels. We hypothesized that this dilemma could be overcome by engineering the strength of the first physical network and the polymer/filler interaction. Similar to our rheological tests, we conducted a series of tensile tests to understand the effects of surface charges on the mechanical strength of the zwitterionic nanocomposite hydrogels. After we cured the hydrogel through the photopolymerization of AA and DMAPS, the hydrogels were further immersed in an aqueous solution of AlCl_3_ for 10 h. During the immersion, the acrylic acid groups in the copolymers were ionically cross-linked by the Al^3+^ ions. Our initial investigation, as indicated in Appendix A, showed that 25 wt.% DMAPS in the mixture of monomers induced the highest toughness of the hydrogels, potentially due to a balanced hydrogen bonding and ionic interaction. This finding is in line with the previous work of Lei et al. [17] who used AA/DMAPS to prepare robust hydrogels. We also found that 10 h was an optimal time for the immersion in the aqueous solution of AlCl_3_ in order to achieve sufficient ionic cross-links and mechanical strength (Appendix A). In addition, immersing in deionized water or in an aqueous solution of NaCl only gave weak hydrogels (Appendix A). However, directly adding AlCl_3_ to the precursor ink would be a better way to control the number of ionic cross-links. We found the dispersion of CCNCs in an aqueous solution of AlCl_3_ was not stable, possibly due to the severe aggregation of the nanocrystals induced by multivalent salts [31].

Zwitterionic nanocomposite hydrogels of CNCs and CCNCs with different filler contents were prepared for tensile tests (Figure 4). The hydrogel without any nanocrystals as fillers could be stretched up to 2400%, but it was a weak gel with a low Young’s modulus of 0.036 MPa. When the concentration of CNCs was gradually increased, the nanocomposite hydrogels became stiffer. At 12 wt.% CNCs, the nanocomposite hydrogels showed a maximum stress of 0.54 MPa and elongation up to 1257%. For CCNCs, the nanocomposite hydrogels also became stiffer when the filler contents increased, but CCNCs were more effective in strengthening the hydrogel. The hydrogel with 7 wt.% CCNCs had a maximum stress of 0.66 MPa and high elongation up to 1190%, even better than the hydrogel using 12 wt.% CNCs. Besides the tensile test, SEM images of the freeze-dried hydrogels (Appendix A) showed hierarchical networks for both nanocomposite hydrogels. Compared to the hydrogels without the nanocrystals, some CNC and CCNC fibrils could be found in the nanocomposite hydrogels. The physical network of the nanocrystals was embedded inside the polymer matrix. Figure 4b,d summarize the effect of filler contents on the Young’s modulus and toughness of the hydrogels prepared with CNCs or CCNCs. CCNC hydrogels showed both higher modulus and toughness than CNC hydrogels. Therefore, even at low concentrations, CCNCs can effectively reinforce the hydrogels and absorb more energies than the hydrogels with CNCs. Previous work on CNC nanocomposite hydrogels often used a chemically cross-linked polymer network with CNCs. Even with the reinforcement of CNCs, these hydrogels are typically soft, with Young’s moduli less than 100 kPa and toughness less than 1000 kJ/m^3^ [41,51,52,53]. For example, Yang et al. [53] fabricated nanocomposite hydrogels with CNCs and a chemically cross-linked poly(acrylamide) network. Their hydrogels only had a low Young’s modulus of 33.8 kPa and an elongation-at-break of 800%. Zhang et al. [40] used CCNCs and an Fe^3+^ cross-linked poly(acrylic acid) network to fabricate an extremely strong hydrogel with a Young’s modulus up to 15.6 MPa and toughness up to 18.9 MJ/m^3^. However, the stretchability of their hydrogels was limited to around 300%. Our CCNC zwitterionic nanocomposite hydrogels demonstrated both high modulus and stretchability for application in wearable sensors.

Besides the tensile strength, we found that the hydrogels with CCNCs dissipated more energies than the hydrogels of CNCs. As shown in Figure 5, we investigated the magnitude of dissipated energies (*U_hys_*) for different samples with loading–unloading tests to 300% strain. For both CNCs and CCNCs, *U_hys_* gradually increased with the filler contents, indicating the fracture of the first physical networks from the nanocrystals. However, at the same filler contents (7 wt.%), the *U_hys_* of nanocomposite hydrogels was 255 kJ/m^3^ for CCNCs and only 173 kJ/m^3^ for CNCs. Our previous results [39] and the work done by Chen et al. [11] have shown that the mechanical strength and the toughness of the double-network hydrogels are strongly related to the stiffness of the first physical network. As shown in our rheological analysis, the ink of CCNCs provided a stronger network than the one prepared with CNCs. During stretching, more energy is required to fracture the CCNC networks due to the strong particle–particle attraction. At low concentrations, CCNCs effectively formed a network for both printing and mechanical reinforcement.

Another important property of these nanocomposite hydrogels is their capability for self-recovery. Due to the fracture of the networks, the nanocomposite hydrogels may be softened after stretching. Fortunately, some of the fractured networks can recover to their initial states to strengthen the hydrogels again. We performed cyclic loading–unloading tests for the zwitterionic nanocomposite hydrogels containing 7 wt.% CCNCs to different maximum strains (Appendix A). The size of the hysteresis loop gradually increased with the maximum strain of each cycle, indicating a hierarchical fracture of the CCNC network. Successive loading–unloading tests for the same hydrogels to different strains without any rest also showed a gradual increase of hysteresis after each cycle.

Furthermore, the nanocomposite hydrogels showed partial self-recovery as each loading curve overlapped with the previous unloading curve. We thus analyzed the time required for self-recovery. The printable hydrogels of CNCs and CCNCs were stretched to 300% strain, unloaded, and loaded again after different waiting times (Figure 6). As discussed earlier, the CCNC nanocomposite hydrogels had higher *U_hys_* than the CNCs hydrogels. Both hydrogels were significantly softened after the first cycle. Only around 50% of *U_hys_* could be recovered if the hydrogels were immediately loaded again. After resting for 1 min, 2 min, and 5 min, both hydrogels could progressively recover to at least 70% of their initial *U_hys_*. Although the CNCs hydrogels recovered to 79.3% of their initial value, higher than the CCNCs hydrogels (70.0%), more energy was required to break the stiff CCNC network. The magnitudes of *U_hys_* recovered by the CNCs or CCNCs at a given resting time were similar. Neither hydrogel could fully recover because of the permanent fracture of CNC or CCNC networks. Nevertheless, the above results indicate that the nanocomposite hydrogels could partially recover to their initial strengths and that the surface functionalization of the nanocrystals had no significant effect on the recovery rate. In addition, the previously reported nanocomposite hydrogels using CCNCs typically had a slow recovery rate and needed more than 1 h of resting time to achieve 70% of their original *U_hys_* after each cycle [40,41]. Therefore, our CCNC nanocomposite hydrogels showed enhanced fatigue resistance, potentially due to the fast rearrangement of the polymer network.

We further attempted to use FT-IR to probe the interaction between the polymers and the nanocrystals (Appendix A). After immersing the hydrogels into the aqueous solution of AlCl_3_, the characteristic peak of C=O stretching shifted slightly from 1706 to 1697 cm^−1^, indicating the structural change of the ionized carboxylic groups with the Al^3+^ ions [54]. We also found two peaks at 1058 and 1036 cm^−1^. These signals were likely from the sulfate half-esters of the nanocrystals. Other than that, we found no significant changes in the spectra for the hydrogels using CNCs or CCNCs. Interestingly, the water contents of both nanocomposite hydrogels were not significantly different (Appendix A). Therefore, the amount of water in the hydrogels was not the main reason for the mechanical toughness of the CCNC nanocomposite hydrogels. According to our previous rheological analysis, the ink using 12 wt.% CNCs had a higher *G’_0_* than that of the 7 wt.% CCNCs ink. However, the toughness and the *U_hys_* of the nanocomposite hydrogels using 12 wt.% CNCs were not significantly larger than those of the CCNC hydrogels. Therefore, the strength of the physical network might not have been the only factor for the reinforcement of the CCNC nanocomposite hydrogels. Although FT-IR could not detect the specific interaction between the polymers and the nanocrystals, the mechanical reinforcement of CCNCs may have come from multiple weak interactions and the strong physical network of CCNCs, as suggested by Ning et al. [55]

### 3.3. Ionic Conductivity of the CCNC Nanocomposite Hydrogels

We performed electrochemical impedance spectroscopy (EIS) to survey the effects of different nanocrystals on the ionic conductivity of the nanocomposite hydrogels. In Figure 7a,b, the hydrogels made by CNCs or CCNCs showed similar trends in the Nyquist plots under various filler contents. The hydrogels with 7 wt.% CCNCs had a bulk resistance of 29 Ω and high ionic conductivity of 0.9 S/m. As summarized in Figure 7c, both types of hydrogels had high ionic conductivity even at relatively high filler contents (7 wt.% for CCNCs and 12 wt.% for CNCs). The concentration of fillers had a minimal effect on the ionic conductivity of our hydrogels. The ionic conductivity of the CCNC hydrogel was slightly higher than that of the CNC hydrogel, possibly because of additional counter ions introduced by the surface modification. Our previous work using deep eutectic solvents with unmodified CNCs showed a significant decrease in ionic conductivity when adding more nanocrystals [39]. However, a much larger amount of non-conductive CNCs was needed to form strong physical networks in the deep eutectic solvents. The high viscosity of the solvent may also have inhibited the movement of ions through the gels. The good ionic conductivity of the nanocomposite hydrogels indicates their potential as wearable sensors.

### 3.4. CCNC Nanocomposite Hydrogels as Wearable Sensors

We then used 3D printing to fabricate a capacitance thin-film sensor. Based on our previous discussion, we chose the ink with 7 wt.% CCNCs because it exhibited good printability and mechanical strength. Following the design in Figure 8a, the ink was printed as thin films, photopolymerized, and then immersed in an aqueous solution of AlCl_3_ for 10 h to properly cross-link the hydrogels. A stretchable VHB tape as dielectric was sandwiched between two CCNC nanocomposite hydrogels. The two hydrogels were encapsulated by extra layers of VHB to prevent the evaporation of water.

The assembled sensor showed good sensitivity to the change of capacitance during mechanical deformation. We first performed a cyclic tensile test to different maximum strains and monitored the change of relative capacitance, ((*C* − *C_0_*)/*C_0_*). *C_0_* was the initial capacitance at 0% strain and *C* was the real-time capacitance of the sensor. In Figure 8b, the capacitive sensor successfully showed distinct and reproducible responses from 1 to 200% strain. The gauge factor (GF = ((*C* − *C_0_*)/*C_0_*)/*ε*) of the sensor was calculated from the results of Figure 8c. The sensor had a linear response of capacitance with its GF equal to 0.67. The sensitivity of the strain sensor was slightly lower than reported values (0.86 to 0.92) [56,57], potentially due to the insufficient adhesion between the hydrogels and the dielectrics. The durability of the sensor was tested by continuously loading and unloading the strain sensor at 30% strain. As shown in Figure 8d, the responses of the sensor remained steady even after 100 cycles.

To further demonstrate the potential of this sensor, we mounted the sensor on a volunteer’s finger and elbow. As shown in Figure 8e,f, the sensor exhibited reproducible responses for both large and small movements of the human body, such as bending the elbow or finger. The above results demonstrate that the printed CCNC nanocomposite hydrogels can successfully act as stretchable ionic conductors for a capacitive strain sensor.

Finally, we used DIW to increase the sensitivity of tactile sensors. As well as their use as strain sensors, capacitive devices are commonly used as tactile sensors that detect pressure changes upon touching or pressing. The sensitivity of the tactile sensors depends on the change of contact area between the hydrogels and the dielectrics. For simple parallel-plate geometry, the sensitivity comes from the intrinsic compressibility of the hydrogels and the dielectrics. Several recent studies [57,58] have used hydrogels with structured surfaces to enhance the compressibility and the sensitivity of the tactile sensors. Here, we printed grooved hydrogels (Figure 9a) for the tactile sensors. Owing to the high yield stress of the CCNC ink, we were able to construct a series of grooves between the contact of the hydrogels and the dielectrics. The depth of each groove was 800 μm and the line-to-line spacing was 400 μm. The detailed designs can be found in Appendix A. The grooved hydrogels were more deformable than simple thin films. As shown in Figure 9b, the grooved sensor had higher gauge factors than the film sensor, especially when only small pressure (<0.2 kPa) was applied. The sensitivity of our device was comparable to previous work [57,58] with structured hydrogels (0.45 kPa^−1^ to 0.84 kPa^−1^). The grooved sensors also had a stable response over 500 cycles of pressing and releasing (Figure 9c). When the grooved sensor was mounted on a volunteer’s hand, it could detect both rapid and slow pressure changes upon pressing (Figure 9d). The above studies show that our CCNC nanocomposite ink has great potential for use in wearable sensors. 3D printing allows a facile design and fabrication for sensors with enhanced sensitivity.

## 4. Discussion

Our findings suggest that the surface modification of CNCs enhances the capability of the nanofiller as a rheological modifier and reinforcement for hydrogels. We anticipate that further engineering on the surface of CNCs may lead to tougher and self-healable nanocomposite hydrogels. For the 3D printing of wearable sensors, we have shown that the printed hydrogels can be assembled into capacitive sensors. Compared to the simple thin-film geometry, the grooved surface of hydrogels fabricated by DIW further increased the sensitivity of the tactile sensors. Similar to the findings of our previous work [39], 3D printing may enable more complex structures in the sensors for anisotropic responses or enhanced sensitivity. Although we only showed one design for grooved hydrogels for tactile sensors, we anticipate that further optimization for the surface pattern of the hydrogels by 3D printing can achieve robust sensors.

## 5. Conclusions

In conclusion, we designed a 3D printable nanocomposite hydrogel with CCNCs that showed high mechanical strength, stretchability, and transparency. By engineering the surface chemistry of the nanocrystals, the CCNCs formed a strong physical network at a concentration lower than for conventional CNCs. The polymerization of AA/DMAPS, and the subsequent ionic cross-linking by Al^3+^, introduced multiple physical networks for the nanocomposite hydrogels. The strong physical network of CCNCs and the polymers/nanocrystals association imparted the high toughness of the hydrogels even when the concentration of the filler was reduced. We further demonstrated the potential of the printed CCNC nanocomposite hydrogels as stretchable ionic conductors for capacitive sensors. 3D printing provides a facile strategy to fabricate surface-structured hydrogels that enhance the sensitivity of the sensors. The assembled sensor displayed good sensitivity, stability, and the capability to monitor different human activities. Our results show that renewable nanomaterials can be used for the integration of additive manufacturing and wearable electronics. A judicious surface modification of CNCs can simultaneously achieve the 3D printability, transparency, and mechanical toughness of the nanocomposite hydrogels.

## Figures and Tables

**Figure 1 polymers-13-00688-f001:**
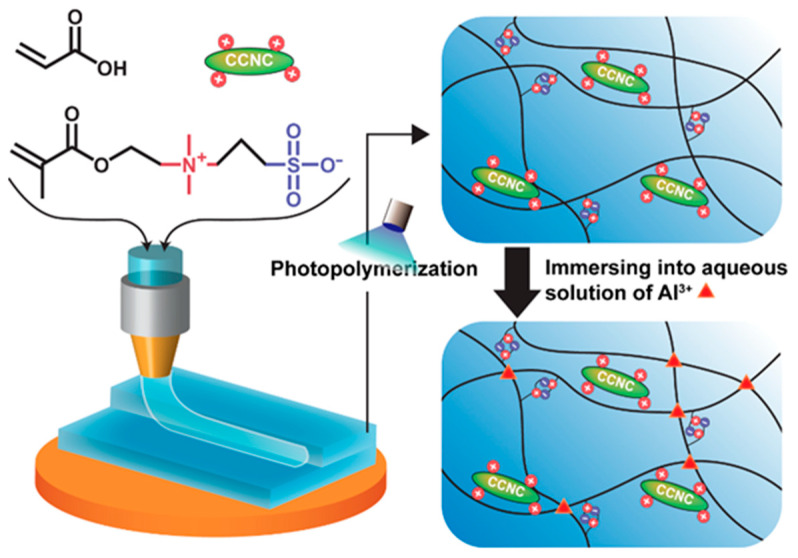
Synthesis and 3D printing of the cationic cellulose nanocrystal (CCNC)-reinforced zwitterionic nanocomposite hydrogels.

**Figure 2 polymers-13-00688-f002:**
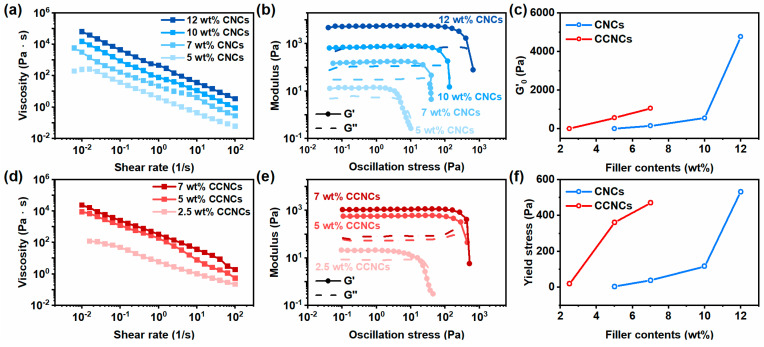
Rheological behavior of the aqueous ink with different concentrations of cellulose nanocrystals (CNCs) and CCNCs. (**a**,**d**) Steady-state shear viscosity of the ink. (**b**,**e**) Oscillatory rheological measurement. (**c**) The dependence of initial storage modulus *G’_0_* on the concentration of CNCs (blue) and CCNCs (red). (**f**) The dependence of yield stress on the concentration of CNCs (blue) and CCNCs (red).

**Figure 3 polymers-13-00688-f003:**
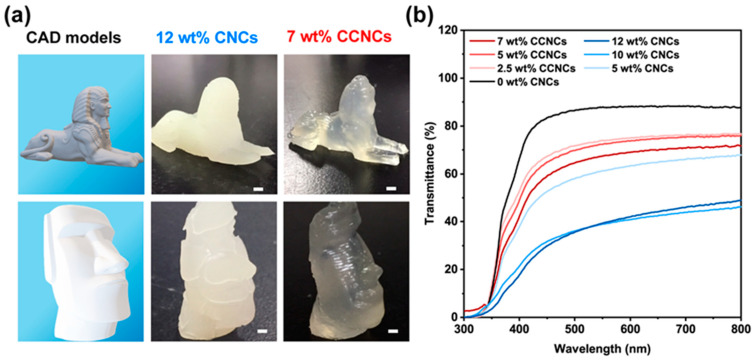
Optical properties and the 3D printed models with different nanocrystal inks. (**a**) 3D printed sphinx and Moai statue. The first column shows the CAD models. The second and third columns show the hydrogels using 12 wt.% CNCs and 7 wt.% CCNCs, respectively. The nozzle diameter was 0.4 mm. The scale bars correspond to 1 cm. (**b**) Transmittance spectra of the printed hydrogel film using various concentrations of CNCs or CCNCs.

**Figure 4 polymers-13-00688-f004:**
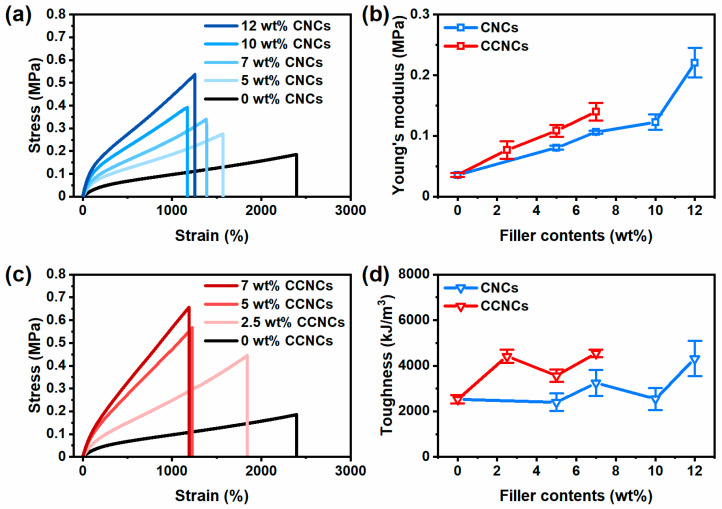
Mechanical properties of the nanocomposite hydrogels. (**a**) Tensile stress–strain curves of the hydrogels with different concentrations of CNCs. (**b**) The effect of different nanocrystals on the Young’s moduli of the nanocomposite hydrogels, CNCs (blue) and CCNCs (red). (**c**) Tensile stress–strain curves of the hydrogels with different concentrations of CCNCs. (**d**) The effect of different nanocrystals on the toughness of the nanocomposite hydrogels, CNCs (blue) and CCNCs (red).

**Figure 5 polymers-13-00688-f005:**
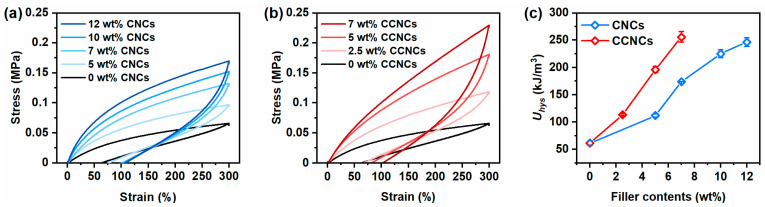
(**a**,**b**) Loading–unloading curves of the nanocomposite hydrogels using different concentrations of CNCs or CCNCs. (**c**) Dissipated energies (*U_hys_*) for CNCs (blue) and CCNCs (red).

**Figure 6 polymers-13-00688-f006:**
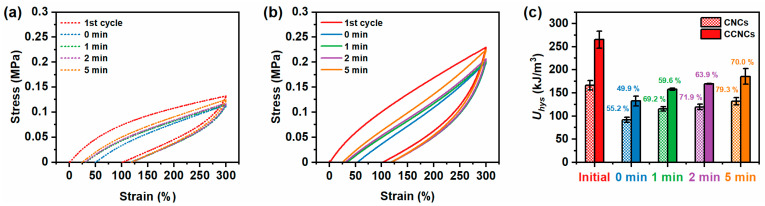
Self-recovery of the nanocomposite hydrogels after different resting times. (**a**) CNC (12 wt.%) nanocomposite hydrogels. (**b**) CCNC (7 wt.%) nanocomposite hydrogels. (**c**) Recovered *U_hys_* at different resting times.

**Figure 7 polymers-13-00688-f007:**
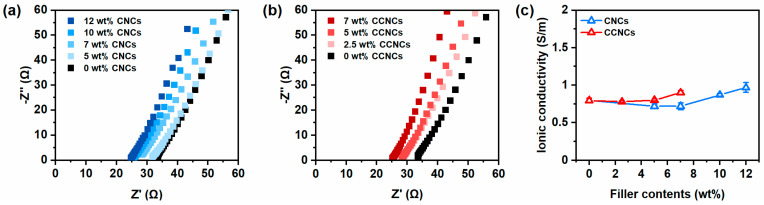
Ionic conductivity of different zwitterionic nanocomposite hydrogels. (**a**) Nyquist plots of the hydrogels with different concentrations of CNCs. (**b**) Nyquist plots of the hydrogels with different concentrations of CCNCs. (**c**) The effect of filler contents on the ionic conductivity, CNCs (blue) and CCNCs (red).

**Figure 8 polymers-13-00688-f008:**
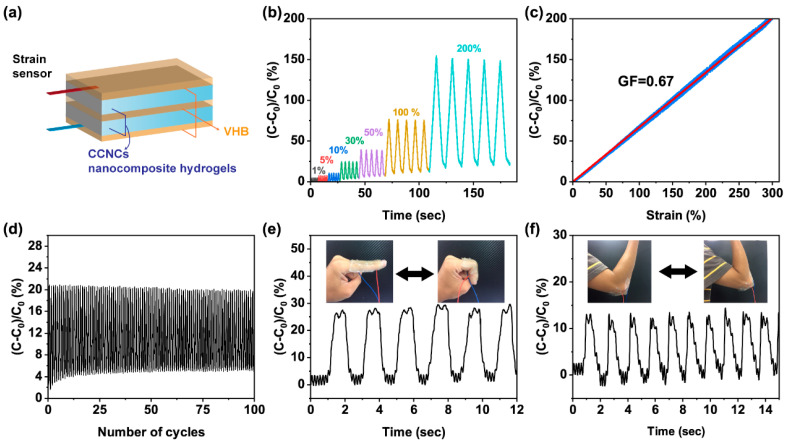
Performance of the 3D printed strain sensor. (**a**) The assembly of the strain sensor. (**b**) Relative capacitance change of the sensor under different strains. (**c**) The dependence of relative capacitance change on the strain. The red line was the linear fitting of the gauge factor in the range of strain from 0 to 300%. The R^2^ value of the linear fitting was 0.99. (**d**) Relative capacitance change of the sensor at 30% strain for 100 cycles. (**e**,**f**) The relative capacitance change when the sensor was attached to the finger and elbow.

**Figure 9 polymers-13-00688-f009:**
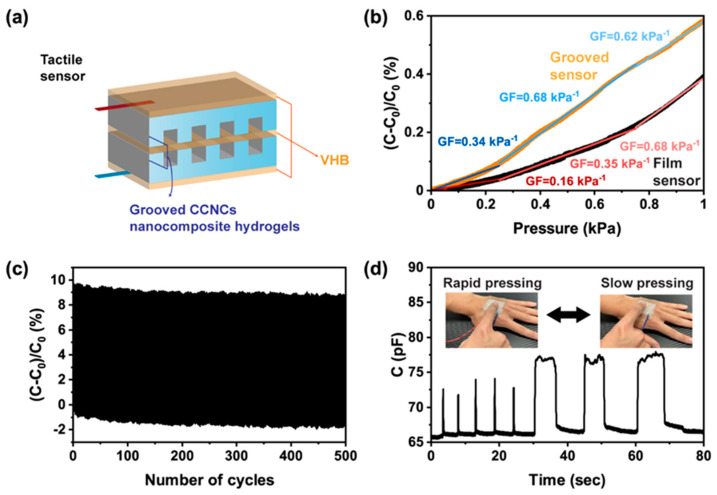
Performance of the 3D printed tactile sensor. (**a**) The design of the grooved sensor. (**b**) Relative capacitance change of the sensors under different pressure. The orange and black lines are the responses of the grooved sensor and the film sensor, respectively. The slope of linear fitting corresponds to gauge factors in the ranges of pressure from 0 to 0.25 kPa, 0.25 to 0.75 kPa, and 0.75 to 1 kPa. The R^2^ value of each linear fitting was above 0.98. (**c**) Relative capacitance change of the grooved sensor at 5% strain for 500 cycles. (**d**) The real-time capacitance of the grooved sensor under different frequencies of pressing by a volunteer.

## Data Availability

Data is contained within the article.

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
