# Peer review of "Cationic Cellulose Nanocrystals-Based Nanocomposite Hydrogels: Achieving 3D Printable Capacitive Sensors with High Transparency and Mechanical Strength"

_polymers, 2021, doi:10.3390/polym13050688_

Round 1

Reviewer 1 Report

In this paper (Polymers-1112966), the authors proposed cationic cellulose nanocrystal (CNC) as a bio-additive for 3D printable nanocomposite hydrogels with high mechanical strengths, stretchability, and transparency. The synthesized 3D printable hydrogel was used for fabrication of strain sensors. The effect of cationization on properties of hydrogel in the introduced manuscript is important. This manuscript is recommended to be published after including and addressing the below listed comments with major corrections.

- A very recent paper (3D printable strain sensors from deep eutectic solvents and cellulose nanocrystals, ACS Appl. Mater. Interfaces 2020, 12, 34235-34244) has been published by the authors. Although, in that paper they utilized CNC not CCNC, however, there are many similarities in this work and previous report from materials and application perspectives. Authors should mention about that results and clearly explain the novelty and importance of this work compared with their previous publication in the introduction section of the manuscript.

-  In line 98, the authors claimed that such hydrogels has not been explored for wearable sensors previously. However, similar wearable sensors were proposed in their previous work.

- The authors mentioned about the diameter of CNC measured by atomic force microscopy (AFM) in Materials section of the manuscript. Authors should include the AFM images of CNC and CCNC in the revised manuscript or supporting information.

- The authors tried to explain the effect of CCNC on rheology by zeta potential value, however, the mechanism of interaction between CNC or CCNC with AA and DMAPS still remains unclear. What kind of interaction exist between these components? The authors must clearly explain it.

 - Y axis labels for Figures 2b and 2e must be changed since both storage and loss moduli are presented.

- In Figure 3a, why the authors compared 12 Wt% CNCs with 7 wt. % CCNC? Since higher concentration of CNC can increase the opacity of hydrogel. It is recommended that the authors compare same concentration of CNC and CCNC in the hydrogel.

- The octopus structure proposed in Figure 3a, was previously shown in the previous paper published by the authors: ACS Appl. Mater. Interfaces 2020, 12, 34235-34244. I recommend authors to eliminate the 3D printed octopus or replace it by other printed structures.

-In line 409, authors mentioned about their previous work. Please write the reference.

- The authors should work on the scientific English of the manuscript and elaborate it. There are some typos in the text. Some expressions and phrases are also awkward.

Reviewer 2 Report

I have reviews the manuscript entitled “Cationic Cellulose Nanocrystals-based Nanocomposite Hydrogels: Achieving 3D Printable Capacitive Sensors with HighTransparency and Mechanical Strength”.

The primary aim of the work is to develop highly thixotropic ink with high capacitive sensing ability by using cationic cellulose nanocrystals. I found this work very interesting and enjoyed reading it.

I think it is appropriate for the publication after minor considerations:

1) It is recommended to employ power-law model to quantify the shear-thing behaviour of the ink as following papers

https://pubs.rsc.org/fi/content/articlehtml/2020/nr/d0nr02581j

https://pubs.rsc.org/en/content/articlehtml/2020/tb/d0tb00627k

2) It is recommended to discuss about other type of cellules such as methylcellulose and justify why you selected CNC.

https://pubs.acs.org/doi/full/10.1021/acsabm.0c00169

https://pubs.rsc.org/en/content/articlelanding/2020/bm/d0bm00027b#!divAbstract

3) Please add more details about the printing parameters such as pressure, printing speed, distance of nozzle tip from platform.

Author Response

Please the attachment.

Round 2

Reviewer 1 Report

Thanks for the revised version, which has materially improved the manuscript. .  I think this is ready to publish.

Reviewer 2 Report

The respected authors addressed well the comments. I think it is suitable for the publication in the present form.